# External Validation of the ‘*PHYT in Dementia’*, a Theoretical Model Promoting Physical Activity in People with Dementia

**DOI:** 10.3390/ijerph17051544

**Published:** 2020-02-28

**Authors:** Claudio Di Lorito, Alessandro Bosco, Kristian Pollock, Rowan H. Harwood, Roshan das Nair, Pip Logan, Sarah Goldberg, Vicky Booth, Kavita Vedhara, Maureen Godfrey, Marianne Dunlop, Veronika van der Wardt

**Affiliations:** 1Division of Rehabilitation, Ageing and Wellbeing, School of Medicine, University of Nottingham, Queen’s Medical Centre, Nottingham NG7 2UH, UK; mczpal@exmail.nottingham.ac.uk (P.L.); mszvb1@exmail.nottingham.ac.uk (V.B.); Maureengodfrey47@gmail.com (M.G.); Mariannedunlop@icloud.com (M.D.); 2Division of Psychiatry & Applied Psychology, School of Medicine, University of Nottingham and Institute of Mental Health, Triumph Road, Nottingham NG7 2TU, UK; alessandro.bosco@nottingham.ac.uk (A.B.); mszrdn@exmail.nottingham.ac.uk (R.d.N.); 3School of Health Sciences, University of Nottingham, Queen’s Medical Centre, Nottingham NG7 2UH, UK; ntzkp@exmail.nottingham.ac.uk (K.P.); ntzrhh@exmail.nottingham.ac.uk (R.H.H.); mczseg@exmail.nottingham.ac.uk (S.G.); 4Division of Primary Care, School of Medicine, Room 1305 Tower Building, University Park, Nottingham NG7 2RD, UK; lwzkv1@exmail.nottingham.ac.uk; 5Wissenschaftliche Mitarbeiterin, Zentrum für Methodenwissenschaften und Gesundheitsforschung Abteilung für Allgemeinmedizin, Präventive und Rehabilitative Medizin, Philipps-Universität Marburg Karl-von-Frisch-Straße 4, 35032 Marburg, Germany; v.vanderwardt@uni-marburg.de

**Keywords:** physical activity, exercise, behaviour change, dementia, theory, external validation

## Abstract

Physical activity is beneficial for people with dementia. We previously developed a theoretical model to explain behaviour change in physical activity in dementia (PHYT-in-dementia). This study aimed to externally validate the model. Validation occurred through the process evaluation of a programme promoting activity in people with dementia (PrAISED 2). Twenty participants with dementia and their carers were interviewed to investigate their experience of the programme. The data were analysed through content analysis. The original constructs of the model were used as initial codes and new codes were generated, if elicited from the data. The constructs were also ranked, based on their frequency in the interviews. All of the original model constructs were validated and two novel constructs created: *‘personal history’* and *‘information/knowledge’*. Certain constructs (e.g., support) were more frequently mentioned than others (e.g., personal beliefs). We suggested modifications and integrated them into a revised model. The PHYT-in-dementia recognised that dementia has an impact on motivation to initiate and maintain behaviour change over time. The model advocates that interventions adopt a more holistic approach than traditional behaviour change strategies. The suggested revisions require further validation to accurately predict behaviour change in physical activity in people with dementia.

## 1. Introduction

Dementia is a neurodegenerative condition, characterised by a cluster of symptoms including memory loss, confusion, mood changes and difficulty with day-to-day tasks [1]. Through progression of symptoms, people with dementia are exposed to a higher risk of falling, potentially resulting in injuries and hospitalisation [2,3,4,5].

Regular physical activity and exercise may help to slow down deterioration in people with dementia, therefore potentially reducing the incidence of falls, injuries and hospital admissions [6,7,8,9,10,11]. Physical exercise is *‘planned, structured and repetitive physical activity’* and physical activity is *‘bodily movement produced by skeletal muscles that results in energy expenditure’* [12]. Physical activity and exercise may also promote a person’s mobility and independence, preserve skills [13,14], improve functional ability [15] and cognitive function [16], and have a positive effect on quality of life and wellbeing [17]. The benefits of exercise may extend to the carers of people with dementia, resulting in experiencing reduced care burden [18,19].

A recent systematic review identified 41 exercise and physical activity interventions designed specifically for people with dementia, including fitness/aerobic exercises, exercises for coordination, balance and flexibility, strength exercises, endurance/resistance training and walking programmes [20]. Despite the abundance of research in this area, there is currently no theoretical framework to inform the design of effective interventions promoting exercise in people with dementia. The existing behaviour change theories may not be applicable to capture the uniqueness of the experience of living with dementia (e.g., memory problems having a negative impact on motivation and confidence) and may fail to accurately identify the factors associated with physical activity and how these mediate adherence and intervention outcomes.

The UK Medical Research Council advocates the use of theory in developing complex interventions [21], as theory allows the identification of the factors impinging on the ability/motivation of the person to fully engage in the intervention and maintain the behaviour over time [22,23]. We previously developed the PHYT-in-dementia (Physical Activity Behaviour change Theoretical model in dementia), explaining behaviour change in physical activity in people with dementia [24]. In our previous work, we sought literature that identifies theories used to explain behaviour change in physical activity in adult populations without a diagnosis of dementia; we extrapolated the theories’ main constructs (i.e., variables identified as mediating behaviour change); we synthesised the constructs (based on commonalities), adapted them to a population with dementia and face-validated them. The end result was the original version of the PHYT-in-dementia (Figure 1).

Empirical data gathered in exercise intervention studies with people with dementia give preliminary support to the relevance of some of the constructs identified in the original version of the PHYT-in-dementia. For example, a recent review of the literature [25] has found that higher motivation to adhere to an intervention programme is linked to certain intervention characteristics, including the use of behaviour change techniques (e.g., motivational interviewing), the provision of tailored supervision/activities to meet participants’ individual needs, the setting of SMART (i.e., specific, measurable, attainable, realistic, time-bound) goals, and the use of booklets/guidance on how to do the exercises. The construct “carer’s characteristics” has also proved relevant in relation to motivation to exercise in people with dementia. It has been found that carers who fear that the health and safety of the person they care for might be compromised through exercise may pose barriers to the person’s motivation to be/get physically active [26,27,28]. “Social opportunity” is another construct that seems to have received validation in the existing literature. Studies [29,30] have found an association between higher motivation to exercise adherence and delivery of exercise classes in a group format (as opposed to individual delivery), suggesting the relevance of aspects including opportunities for socialisation.

However, further validation (i.e., testing models with data) is recognised as crucial to determine the credibility of a model and to correct its parameters [31]. The International Society for Pharmaco-economics and Outcomes Research (ISPOR) and the Society for Medical Decision Making (SMDM) Modeling Good Research Practices Task Force-7 [32] contend that external validation (comparing model results to real-world event data) is critical. The aim of this study was to externally validate and propose a revised version of the model.

## 2. Materials and Methods

This study followed guidance by the Modeling Good Research Practices Task Force-7 [32]. The guidance prescribes that validation should be set up to match real-life scenarios as closely as possible, including setting, target populations, and treatment [32]. We conducted the validation study in the context of the process evaluation of the promoting activity, independence and stability in early dementia (PrAISED) [33].

PrAISED is a multicentre, individually-randomised, pragmatic, parallel-group, controlled trial, testing the clinical and cost-effectiveness of a therapy intervention including exercises, activities of daily living and dual tasks (physical and cognitive exercises). Three hundred sixty-eight participants were recruited to take part in PrAISED through memory clinics, general practice registers, dementia support groups and the National Institute for Health Research (NIHR) Join Dementia Research register. The participants were randomised to one of two arms: the intervention arm, comprising therapy intervention delivered in the participants’ homes by trained therapists, or the control arm, comprising treatment as usual (i.e., falls risk assessment and advice) [34].

The process evaluation sub-study investigated *‘How PrAISED works’* through a mixed-methods design. The data used to validate the PHYT-in-dementia were obtained from the semi-structured qualitative interviews the PrAISED participants took part in, to identify, among several mechanism of impact, the constructs (or variables) mediating motivation to engage in physical activity and exercise.

The PrAISED 2 trial has received ethical approval number 18/YH/0059. The ISRCTN Registration Number is 15320670. The trial sponsor is Research and Innovation, Nottingham University Hospitals NHS Trust. The PrAISED ethical approval includes the process evaluation work package, upon whose data this study is based. An independent Programme Steering committee (PSC) and Data Monitoring and Ethics Committee (DMEC) meet six-monthly to monitor progress and ethical compliance in accordance with NIHR procedures.

### 2.1. Participants and Setting

To follow best practice in external validation [32], we purposively recruited participants with dementia and their respective identified care giver (henceforth defined ‘carer’) taking part in PrAISED (for inclusion criteria, see protocol of PrAISED [34]). By doing so, we aimed to ensure that they reflected the diversity of the PrAISED participants in relation to gender, ethnicity, relationship status, geographical location, and adherence to the exercise programme (i.e., low and high adherence). The participants with dementia were asked whether they preferred to be interviewed alone or together with their carer. All opted for the latter option. The participants with dementia and their carers (henceforth both defined *‘participants’*) were interviewed in their private home in month 6 of the 12-month intervention period of PrAISED.

### 2.2. Data Collection

Once identified, the participants were contacted via telephone by a member of the research team (CDL), who enquired about their willingness to participate in the study. A date for the interview at the participant’s home was then set. On the day of the interview, informed consent was obtained by participants. The qualitative interviews included open-ended questions and prompts (further information on the topic guide is available in the Protocol of the Process Evaluation [33]) and were audio-recorded. They continued until conceptual density was achieved [35] and carried out from April to August 2019.

### 2.3. Data Analysis

The interviews were transcribed verbatim by a professional transcription agency and passed on to the research team, who anonymised quotes by assigning each participants an unidentifiable code (P01, P02 for participants with dementia; C01, C02 for carers). The transcripts were then transferred onto NVivo 12.1 [36].

The data were analysed through content analysis, a systematic coding and categorising strategy to determine patterns and frequency of words used in the transcripts [37,38,39]. We used manifest content analysis (i.e., analysing for the appearance of a word or content in textual material) [37] to test whether the constructs elicited from the transcripts reflected the constructs identified in the original version of the PHYT-in-dementia (e.g., social opportunity).

The original constructs of PHYT-in-dementia and their operational definitions were used as initial constructs (deductive approach). Each response from the participants to the interview questions represented a discrete segment, which was categorised by one author (CDL) into one or more of the constructs. For example, if a participant stated, *“I would not do the exercises, if my wife did not remind me to do them”*, CDL would categorise this segment into the constructs *“The carer”* and *“Support”*. If the segment could be categorised into the initial constructs (i.e., identified in the original model) as in the example above, we did not consider revising the constructs and gave them the status of *‘solidifying constructs’.* If none of the segments could be categorised into the initial constructs, these would be given the status of *‘unclear constructs’.* If new constructs (i.e., not the initial constructs) were elicited from the segments, they were inductively generated and labelled as *‘emerging constructs’*.

The coding and construct validation process was carried out by CDL for all transcripts. However, to contribute perspective and reliability in the coding process, two patient and public involvement (PPI) co-authors with experience of caring for someone with dementia (MG and MD) independently coded 50% of the transcripts (n = 10) each through the same procedure used by CDL. Any disagreement between the PPI raters and CDL (e.g., if a construct was identified as “solidifying” by CDL and “unclear” by the PPI rater) was resolved through consensus in a three-hour meeting between the three raters. This session was also instrumental in carrying out construct refinement. For example, the operational definition of the construct *‘capability’* was expanded to include, as per MG’s PPI lived experience, *‘chronic conditions, such as arthritis, heart problems and stroke, that might compound upon and mitigate against behaviour change*’.

This process resulted in a final codebook, listing the constructs and their operational definitions. To ensure extra construct validity, the codebook was used by a fourth rater (AB) to code two randomly selected interview transcripts (10% of the total). Inter-rater reliability was calculated against CDL’s coding and tested through Cohen’s Kappa coefficient [40] and parameters by Landis and Koch [41]. Where the fourth rater was in disagreement with CDL’s segment categorisation, this was resolved in a team meeting by seeking unanimous consensus across the fourth raters.

Manifest content analysis was also used for a quantitative analysis of the transcripts [42], counting the frequency of the model constructs (i.e., in how many interviews the construct was referred to by the participants) across the interviews. This calculation was used to determine patterns in the constructs (i.e., how strong, recurring they were and how they were distributed across the interviews).

### 2.4. Ethic Approval

The PrAISED 2 trial has received ethical approval number 18/YH/0059. The ISRCTN Registration Number for PrAISED 2 is 15320670.

## 3. Results

The study recruited 40 participants: 20 with dementia and 20 carers. We had 10 participants from each PrAISED research site: Nottinghamshire, Lincolnshire, Derbyshire and Somerset. Thirty-eight participants identified as being of White ethnicity and two identified as being Black. Twenty-two participants were female. Seventeen carers (all spouses) lived with the participants with dementia. Two carers were adult children and one was a sibling of the participant with dementia and lived independently (Table 1). The qualitative interviews lasted 48 minutes on average (range = 31–65).

We found substantial inter-rater reliability (*k*=0.65) between CDL and AB. The analysis of transcripts confirmed the relevance and validity of the original model constructs. Minor changes to terminology were made, which are reported in the constructs’ section below. Two novel constructs were elicited from the study: *‘personal history’* and *‘information/knowledge’.*
Table 2 lists all the constructs with their operational definitions and examples of they may mediate behaviour change. The frequency of the constructs (i.e., in how many interviews the construct was referred to) is illustrated in Figure 2. Below are results from the content analysis, reported by frequency.

### 3.1. Characteristics of the Person with Dementia

The most frequently discussed construct identified through the content analysis was *‘personal characteristics’*. There were certain personality traits linked to behaviour change. Perseverance made the person resilient in the face of adversity:


*“I always try to put my heart and soul into it, I really do. And if I can’t do something, I say ‘oh I’ll give it a rest today and then start again tomorrow’.” (P13, male).*


A sense of competitiveness with oneself or others also promoted behaviour change:


*“I don’t let the thing beat me. One of the ladies (i.e., therapist) when we do the exercises said that I was very competitive.” (P16, male).*


People who enjoyed sports also showed an attitude of readiness to challenge themselves, which was linked to higher engagement in and motivation for behaviour change:


*“I think every time you exercise, there’s always a notch to go one more.” (P02, male).*


On the contrary, participants with a sedentary lifestyle struggled to independently initiate and maintain physical activity:


*“I’m lazy, which is not really helpful, because it’s easier for me to either let somebody else make a decision or not to do something at all.” (P05, male).*


### 3.2. Support

The second most frequently discussed construct was *‘support’.* In the context of dementia, most participants felt that receiving help from significant others was pivotal for their sense of independence:


*“The bus would take my scooter, so I want to try that, but I want somebody to come with me the first time to do it.” (P18, female).*


Support also proved crucial to compensate for memory loss. The participants often relied on significant others to be reminded to do the physical exercises. However, the carer could not always offer the extent of support some participants needed, given their age and deteriorating physical condition:


*“If I fall down, I’ve got to have somebody to lift me up. My wife has problems.” (P12, male).*


As a result, arrangements were in place, so that the participants could count on the extra support of trusted people. Support from the professionals (e.g., therapists delivering the physical exercise programme) was also crucial to enable the person to fully participate in exercise programmes. We found that the participants responded more positively to professionals’ encouragement to exercise than to family members:


*“If the support worker weren’t there, I would try and encourage you (i.e., the participant) to do things, but I don’t think you would do them half as willingly if it was just me.” (C10, female).*


### 3.3. Expectations/Goals

The third most frequently discussed construct was *‘expectations/goals’*. We expanded the original construct *“expectations”* to comprise also the construct of *“goals”*, as most participants reported engaging in behaviour change to obtain specific goals, such as improving mobility and independence:


*“The overriding intention is for my legs to work better than they are; and anything that’s going to help me achieve that, good for me.” (P07, male).*


Others engaged in behaviour change for ‘altruistic’ motives:


*“I want to keep doing it is because I love my family, and we love to go and visit, and that’s it.” (P13, male).*


Acknowledgment of the consequences of not doing activity on both physical and mental health was also a trigger for behaviour change:


*“I think I've got the common sense to do it. Otherwise I’ll just turn into a rigid thing of the past.” (P17, male).*


High expectations on the outcome of behaviour change might disappoint some participants:


*“I said to the doctor ‘I’m having difficulty communicating’ and he said ‘well do you want to try this?’ I said ‘yeah’. And I did. But I was really disappointed because it couldn’t help me with speech.” (P11, male).*


### 3.4. The Carer(s)

The fourth most frequently discussed construct was *‘carer’*. The support from the carer was pivotal in initiating and maintaining behaviour change. However, several carers exhibited high levels of anxiety related to physical activity:


*“I don’t like him being out too long, and I’m sort of looking at my watch to see what time it was when he went.” (C03, female).*


Worries about the potential consequences of falls and injuries might lead some carers to gate-keep activities:


*“Sometimes he wants to do things, but we always say ‘no you can’t do that, you know, it’s too much’.” (C12, female).*


Others, instead, appeared strongly motivated from the get-go to encourage the person they cared for to exercise believing that an element of controlled risk was required to ensure improvement in the person’s condition:


*“Some things can’t be helped. If mum goes out and mum falls, it doesn’t mean that mum should stop going out because there’s always that danger that she falls.” (C19, female).*


Another carer-related aspect that facilitated or limited the person’s ability to engage in physical exercise was rapport with the professionals having daily contact with the person they cared for:


*“I’m not very good with physios anyway, they make you do things you don’t want to do.” (C14, female).*


### 3.5. Progress

The fourth most frequently discussed construct was *‘progress’*. Perception of progress was found to be a highly motivating factor in maintaining behaviour change:


*“When I first started it (i.e., PrAISED), you wouldn’t have recognised me now. I had a big operation, so my body was weak. These exercises encouraged me to build my strength back up.” (P04, male).*


Given its importance as a motivating factor, the participants actively monitored progress:


*“The therapist didn’t say a number of paces that I’m supposed to be doing, but I’m just monitoring to see if it increases or decreases.” (P01, male).*


Incidents could halt progress, and this could have a major impact on behaviour maintenance:


*“That (i.e., progress) is gone out the window now, because although he was getting a bit better, he’s gone backwards with his balance.” (P15, male).*


Conversely, participants reported that once they had fulfilled their potential for improvement, progress would inevitably stop, thus demotivating the person to keep going:


*“We’ve reached stalemate as what I can do physically. That’s the problem isn’t it? So, if there is any more exercises that would be good, but I don’t know what they would be.” (P18, female).*


### 3.6. Social Opportunity

The fifth most frequently discussed construct was *‘social opportunity’*. The social opportunities presented by engaging in behaviour change represented a strong motivator for some participants:


*“It gets me out of the house at least once a week to see a different set of people. That can only be good.” (P01, male).*


This opportunity was especially treasured by participants who previously had an extensive social network:


*“All of mum’s jobs in the past have always been with people, so she’s always had that sort of social interaction. So, she likes that.” (C03, female).*


A need for social inclusion was frequently reported by participants who, because of their age or geographical isolation, had lost social contacts. Therefore, the prospect of potentially reconnecting with old friends strongly encouraged participants to make plans to become active again:


*“I think if I do start swimming again that would make a difference. Because I will be meeting people that I used to meet quite a while ago.” (P05, male).*


However, for less sociable participants or those who were embarrassed to exercise in front of others, social opportunities acted as an inhibitor for behaviour change:


*“I’m quite happy to do those exercises on my own. I’m not that sort of person to join a leisure centre.” (P20, male).*


### 3.7. Self-Efficacy

The sixth most frequently discussed construct was ‘*self-efficacy’*. Dementia could greatly decrease participants’ confidence. In a familiar environment, most participants felt safe, but in novel situations, a sudden loss of a sense of safety made the participant unconfident:


*“If we’re going somewhere different, I think ‘well, how we getting there, how we getting back, what it’s going to be like, who is going to be there?’ And I get a bit jittery.” (P03, male).*


We observed that some participants could struggle to accept the deterioration caused by dementia, thus failing to gather the confidence required to engage in behaviour change:


*“I want to get out to communicate. And I can't because I can't get the words out. So, I'm a bit embarrassed going out with blokes.” (P13, male).*


It was observed that significant others could be instrumental in helping the participants build confidence:


*“I think that was the support worker that said you need the confidence to do it, and she has instilled that confidence in us.” (C04, female).*


### 3.8. Capability

The seventh most frequently discussed construct was *‘capability’*. The participants reported several physical ailments, which limited their physical capability. The impact of physical impairment on motivation to change behaviour led some participants to consider giving up any effort to change behaviour:


*“I was wondering whether I’d give up altogether because I’ve been to three different doctors for my leg and they can’t do anything with it. Well, it’s a bit pointless isn’t it, my trying to strengthen something that cannot be fixed.” (P04, male).*


Limited capability had a negative impact on behaviour change, even in the presence of a strong intrinsic motivation:


*“I always think I could do more than I’m doing. I feel as if I can, but my poor old legs have not responded in the way I would like them to have responded,” (P17, male).*


Capability was thwarted by other factors than mobility. Chronic or acute pain was reported by several participants:


*“One of the earlier ladies was saying that this sitting up one would be good, but of course you can’t actually do that right now though. Your knee is so painful, isn’t it?” (C15, female).*


Prescribed medications could limit capability through side effects:


*“Because he’s not been well for this last fortnight, they are trying to decide whether he’s got Parkinson’s as well. And they prescribed this tablet that’s making him like that.” (C03, female).*


### 3.9. Ideas around the Activity/Intervention

The eighth construct identified through the content analysis was ‘*ideas around the activity/intervention*’. The construct had previously been identified as ‘*characteristics of the intervention’* and changed to reflect the fact it is the subjective views on the activity or intervention, as opposed to the objective features of it, that impinges on motivation to exercise. The participants reported how important tailoring of the intervention were to ensure their full compliance:


*“When the therapist came here, they tried to find out what I liked to do and base it around that.” (P01, male).*


Enjoyment derived from the activity also acted as a strong motivator. The participants enjoyed an activity that required physical exertion, but was still within their realistic level of capability:


*“When I did the exercise, it was about fun, seeing if I could do this. It was a bit of a challenge and that’s great, I enjoyed the session.” (P05, male).*


Another appreciated aspect is delivery of the intervention by trained and experienced professionals:


*“If I was doing the exercise with him, he would not feel as confident as when the therapist is, because of her expertise.” (C09, female).*


Some participants felt that an established routine of exercises, in the context of memory impairment, brought added value to the intervention:


*“They’re the same thing all the time, so I don’t have to learn anything new, and that’s the good side of it.” (P16, male).*


Other, instead, reported that adaptation of the intervention over time was a quality that promoted behaviour change:


*“I did the exercises, and then he realised I was finding it very easy, so they started adding the counting down of numbers while I was doing the exercises.” (P20, male).*


### 3.10. Autonomy/Control

The ninth most frequently discussed construct was ‘*autonomy/control’*. The interviews evidenced how the participants with dementia valued autonomy and independence in their lives:


*“At present I want to make sure that I am as reasonably fit, so that I can do the shopping, I can get out, I can walk around. I don’t want to be trapped in here.” (P08, male).*


Although they relied on the carer’s decisional support, the participants frequently asserted their right to autonomous decision-making:


*“I like to do things when I can just get up and do them, rather than be, well, it’s ten o’clock now, get out there and do a mile. I’ll do the mile when I want to do it.” (P11, male).*


For this reason, the participants really appreciated when others were respectful, and they involved them in the decision-making process:


*“She won’t just say ‘come on, let’s go for a walk’; she’ll always say ‘do you fancy a walk today?’.” (P04, male).*


### 3.11. Physical Infrastructure

The tenth most frequently discussed construct was *‘physical infrastructure’*. For most participants, their home was the most accessible environment in which to engage in behaviour change. In the home, the participants used different strategies that would facilitate physical activity:


*“I usually make sure I’ve got a chair close by, I can put my hand there just in case I felt a bit wobbly.” (P03, male).*


They recognised, however, that exercising alone at home might not be motivating:


*“Personally, I feel that if you’re at home you’ve got to be a lot more self-motivating.” (P12, male).*


In addition, some homes could not fully accommodate the participants’ needs for mobility. Therefore, the participants were happy to venture outdoors:


*“I can do my bending and my feet movement, but the house is a bit small for anything more. I can walk, go straight out, and I can go straight round the field and so on.” (P13, male).*


Characteristics of the outdoors, therefore, also became key to behaviour change. Proximity of outdoor spaces to the home was one of these:


*“I do a bit of rugby just in the field down there. I’m surrounded by nature, which is always very good, because I can do everything in the open air.” (P17, male).*


Very crowded or loud environments could instead be problematic:


*“The trouble is that in the gym it’s very loud. Well, it’s loud that you can’t block it out.” (P12, male).*


### 3.12. Personal History

The eleventh most frequently discussed construct was ‘*personal history’*. The construct had not been identified in the original model. Having been involved in sports prompted engagement with the intervention:


*“Being a PE teacher, swimming has always been a part my life.” (P06, male).*


Participants who had a personal history of being active were usually surrounded by like-minded people, who reinforced their good habits:


*“You generally have friends that are very similar…Football or whatever. You meet that crowd and you stay with them.” (P13, male).*


They were also helped by the fact that they were usually fitter than more sedentary participants:


*“His balance has always been quite good. I think a lot of it is to do with dancing, he’s always danced a lot.” (C10, female).*


A previous history of physical injury, instead, usually had a negative impact on enablement or intention to engage in behaviour change:


*“I had a knee replacement and then I just stopped swimming.” (P17, male).*


### 3.13. Information/Knowledge

The twelfth most frequently discussed construct was ‘*information/knowledge’*. The construct had not been identified in the original model. The necessary information to engage in behaviour change came through different sources. The professionals were a valued source, given their expertise:


*“I’ve got to listen to what you’ve (i.e., the therapists) got to tell me, because you’ve had experience with not just me, but hundreds of other people.” (P12, male).*


The participants were at times disappointed with information (or lack thereof) provided by primary care services:


*“I think that the National Health Service could actually have greater impact. I didn’t actually get any real advice when I was diagnosed.” (P18, female).*


Information was best understood and retained when it was explained in plain language:


*“Once she said to me ‘don’t take no exercise on your own, make sure somebody is in the yard with you’, so she explained plainly to me.” (P10, male).*


### 3.14. The Professional(s)

The thirteenth most frequently discussed construct was ‘*the professional*’. This construct had been previously identified as *“the clinician”*, but the terminology was changed to reflect that any professional (e.g., gym instructor, paid carer, volunteer) being present in the person’s life may have a significant impact on their motivation to be physically active. In the final model, the construct was further collapsed together with the construct *“the carer”* into the category of *“significant other(s), if present”*, to reflect that not every person with dementia has contact with carers and/or professionals.

The study found that the main characteristic of the professional that motivated the participants was their professionalism:


*“I mean when the physio came out recently I thought, you know, she was always, she was spot on.” (P01, male).*


The carers trusted the professionals as a reliable source of information:


*“She even showed us one day what to do if there’s a fall. I had no idea. He’s fallen before, and I haven’t been able to pull him up!” (C03, female).*


The participants reported the importance of the professional’s personal characteristics, other than their technical expertise. An ability and willingness to be patient and actively listen to the person was crucial:


*“She listens to what you say to her. And she seems interested in what I think. I can tell that she is interested and she likes to be part of it all.” (P12, male).*


Showing respect for the person’s wishes and ideas was also highly valued:


*“They just accept what mum tells them, whether they sort of agree with it or not. They’re respectful of mum’s wishes.” (C19, female).*


The participants also appreciated when the professional was able to motivate them to keep active:


*“When she’s gone, I’ve felt very positive about things. She makes you feel as if you can do this and this really gets you started and motivates you.” (P18, female).*


### 3.15. Personal Beliefs

The fourteenth most frequently discussed construct was ‘*personal beliefs’*. Receiving a diagnosis of dementia cast a label on the person that, as a result, developed a pre-conceived system of beliefs about the condition:


*“There’s that danger, you know when people are diagnosed with something, they think things are always worse than what they really are.” (P02, female).*


At times, beliefs on oneself did not necessarily reflect the actual condition of the participant, as reported by this carer:


*“Mum keeps telling me about how bad her memory is, but if I say to mum “we’ve got to do this”, mum will remember.” (C19, female).*


Nonetheless, beliefs about the inevitable deterioration caused by dementia generated high levels of anxiety:


*“It makes me feel so anxious to see whether I’ll be able to swim a length, seeing that I was doing 50 lengths before!” (P15, male)”*


These may also be linked to becoming risk-averse, thus limiting willingness to engage in behaviour change:


*“I don’t do them because I’m afraid I’ll fall over. What if I fall over and can’t get up?!” (P19, male).*


## 4. Discussion

This study aimed to empirically test the external validity and to calibrate the PHYT-in-dementia theoretical model, explaining behaviour change in physical activity in dementia.

The data provided support for all the previously identified constructs of the PHYT-in-dementia, thus giving them the status of *‘solidifying constructs’*. No constructs were therefore rated as ‘*unclear constructs’*. The data also identified two *‘emerging constructs’*: ‘*personal history*’ and ‘*information/knowledge’,* which have been integrated in the suggested revision of the model (Figure 3).

Behaviour change (or behaviour maintenance) occurs within ecological systems, as theorised by the ecological systems theory [43,44]. In this work, we have identified those aspects (i.e., constructs) in the ecological systems that facilitate/thwart motivation to engage in behaviour change. The relevance of some of these, such as support, expectations/goals and information had been previously identified [45]. Results from the study also identified the relevance of personal history, capability, self-efficacy, personal beliefs, personal characteristics, activity/intervention characteristics, physical infrastructure, social opportunity, progress; and autonomy/control. We observe that the constructs might be interrelated and influence each other. For example, the level of support that the person receives to engage in physical activity might affect their self-efficacy. It is also important to emphasise the uniqueness of every individual and that, as a result, a different “dosage” of constructs applies to different people with dementia. For example, the social opportunity construct might be more relevant for behaviour change in sociable individuals.

Motivation (and in turn behaviour change or maintenance) is also dependent on the interaction of the person with significant others, such as the carer(s) and any professional supporting the person to change their behaviour (e.g., gym instructors, trainers, occupational therapists, physiotherapists, support workers). The model recognises that, particularly in the context of dementia, where the person relies on the support of significant others, their characteristics (e.g., views, beliefs, behaviours, attitudes, expectations/goals, support capacity, capability, professionalism) largely affect the person’s success to engage in behaviour change. As shown in the model (through a two-way arrows), the person and the significant others influence each other’s attitudes and behaviours, which can in turn promote/thwart behaviour change. For example, a carer might have risk-averse attitudes toward physical exercise in dementia and this might affect the confidence of the person with dementia, who in turn, might become reluctant to engage in physical activity. Likewise, the person with dementia might challenge the carer’s risk-averse attitude by showing that they are cautious and they know their limits when engaging in physical activity. As a result, the carer might become more supportive of the person’s engagement in physical activity. Figure 4 reports two case-scenarios from PrAISED participants, illustrating how the model constructs apply and interact in particular cases, having a direct impact on the participant’s motivation to engage/maintain behaviour change in physical activity. 

This work is characterised by certain strengths and limitations. It represents the first effort to develop and empirically validate a theoretical model designed to explain the unique experience of behaviour change in the context of physical activity in dementia. The model has undergone different stages of refinement. It has a robust theoretical foundation, being developed through a synthesis of the most relevant theoretical frameworks used to explain behaviour change. In addition, it has been adapted, through several iterations, to a sample with dementia through empirical data and input from patient and public representatives. This has ensured that the model reflects the views and experience of the primary stakeholders and is relevant to dementia.

One limitation of this study is related to generalisability of findings. The model has been only validated in the context of an interview study. In addition, given that the participants were sampled from one study programme (PrAISED), they shared the experience of receiving the same intervention. They also had similar ethnic background. Only three participants with dementia were female. To ensure that the suggested revision of this model truly reflects the diversity of the population with dementia, it would be crucial to undertake multiple validation studies involving a wider range of populations that the model is intended for. For example, given the impact that culture can have on uptake and long-term adherence to interventions, it would be relevant to involve in future studies participants from ethnic minority groups. As recommended by the ISPOR and SMDM Modeling Good Research Practices Task Force-7 [32], it is also crucial, to further validate the PHYT-in-dementia, to test its predictive validity. This process entails recording predicted outcomes based on the model assumptions (e.g., more cognitively impaired participants will adhere less to a physical exercise intervention), waiting for events to unfold (e.g., recording cognitive impairment scores and adherence rates in the context of a randomised controlled trial), and comparing predicted outcomes with observed events. We aim to undertake a predictive validity testing of the model with data available from the PrAISED randomised controlled trial, once they are available.

This novel work adds important findings to the field of behaviour change theories, which can have implications for future practice. Contrary to previous theoretical models, the PHYT-in-dementia emphasises the role of significant others in behaviour initiation and maintenance. The participants to this study often reported that an essential condition for behaviour change was carers reminding, encouraging and motivating participants to exercise, especially as the condition progressed over time. Motivational theories such as the self-determination theory [46] contend that the people will maintain behaviour over time only in the presence of intrinsic motivation. However, in line with previous research recognising the role of carer in the advanced stages of dementia [47], we found that the person might lack the insight/capacity needed to develop and sustain intrinsic motivation, or they might forget that they are intrinsically motivated to change behaviour. As external motivation becomes pivotal [48], the level of commitment that the significant others have in supporting the person with dementia is essential. Previous studies [26,27,28] found that carers might be risk-averse and as a result, adopt gate-keeping behaviours to physical activity. We therefore advocate the importance of providing motivational strategies for carers, incorporating risk enablement and emotional/psychological support to challenge anxieties about falls and injuries (e.g., cognitive behavioural therapy to address negative automatic thoughts).

Another important finding is that the input of non-professional carers alone (e.g., spouses, children) might not be sufficient to motivate the person with dementia, who might struggle to accept suggestions that do not come from trusted (i.e., expert) sources. Our findings showed that the professionals having contact with the person with dementia (i.e., physiotherapists, occupational therapists, and rehabilitation support workers) had a central role in generating motivation to change behaviour. In observing that motivating participants required great skills, and that professionals working with people with dementia might not always possess background knowledge in motivational strategies/techniques, training should be provided. This work can present transferable information to develop effective training for professionals.

## 5. Conclusions

The PHYT-in-dementia recognised the uniqueness of the experience of living with dementia having an impact on motivation to initiate and maintain behaviour change over time. This model advocates that intervention developers and implementers adopt a more holistic approach than traditional behaviour change strategies, where relevant constructs identified through the PHYT-in-dementia are factored in as active agents supporting the person in changing their behaviour.

## Figures and Tables

**Figure 1 ijerph-17-01544-f001:**
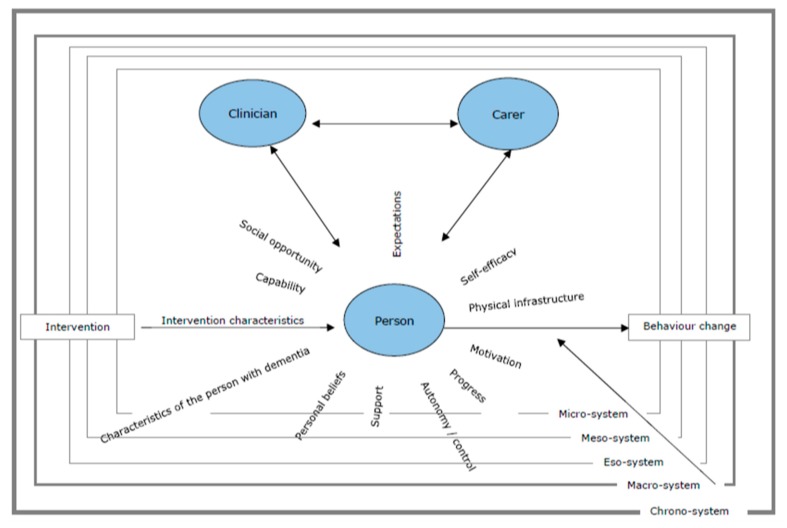
The original PHYT-in-dementia model [24].

**Figure 2 ijerph-17-01544-f002:**
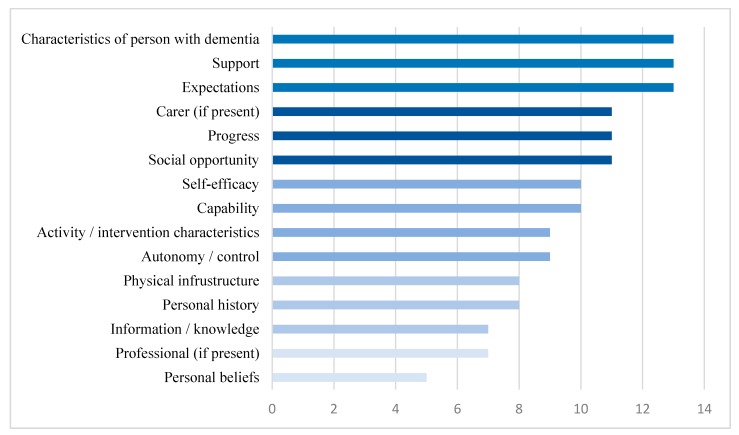
Frequency of constructs in the interviews (n).

**Figure 3 ijerph-17-01544-f003:**
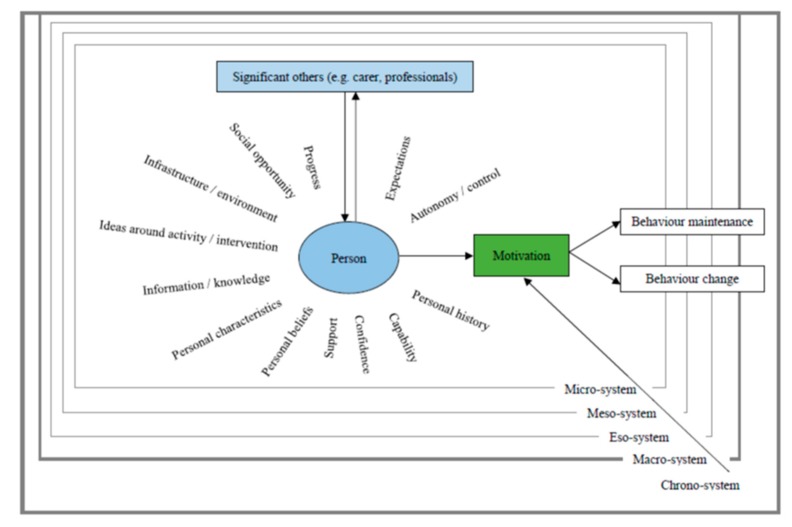
Proposed revision of the PHYT-in-dementia.

**Figure 4 ijerph-17-01544-f004:**
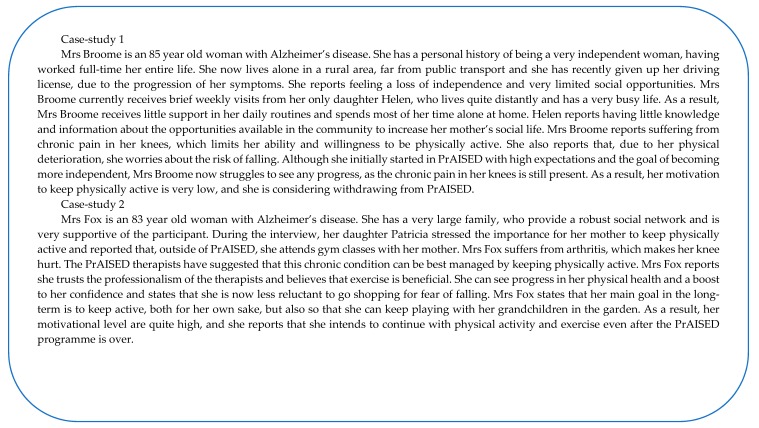
Two case-studies form promoting activity, independence and stability in early dementia (PrAISED) participants, showing application of the PHYT-in-dementia model constructs.

**Table 1 ijerph-17-01544-t001:** Characteristics of participants.

*Participant ID **	*Gender*	*Age*	*Ethnicity*	*Living Arrangement*	*Relationship to Carer*
*P01*	M	75	White	Lives independently	Brother
*P02*	M	83	White	Lives with carer	Spouse
*P03*	M	76	White	Lives with carer	Spouse
*P04*	M	73	White	Lives with carer	Spouse
*P05*	M	90	White	Lives with carer	Spouse
*P06*	M	80	White	Lives with carer	Spouse
*P07*	M	78	White	Lives with carer	Spouse
*P08*	M	83	White	Lives with carer	Spouse
*P09*	M	75	White	Lives with carer	Spouse
*P10*	M	70	White	Lives with carer	Spouse
*P11*	M	77	White	Lives with carer	Spouse
*P12*	M	85	White	Lives with carer	Spouse
*P13*	M	81	White	Lives with carer	Spouse
*P14*	M	80	White	Lives with carer	Spouse
*P15*	M	85	White	Lives with carer	Spouse
*P16*	M	74	White	Lives with carer	Spouse
*P17*	M	77	White	Lives with carer	Spouse
*P18*	F	85	White	Lives independently	Mother
*P19*	F	83	Black	Lives independently	Mother
*P20*	M	86	White	Lives with carer	Spouse
*C01*	F	80	White		
*C02*	F	81	White		
*C03*	F	73	White		
*C04*	F	73	White		
*C05*	F	88	White		
*C06*	F	72	White		
*C07*	F	75	White		
*C08*	F	82	White		
*C09*	F	76	White		
*C10*	F	70	White		
*C11*	F	71	White		
*C12*	F	72	White		
*C13*	F	78	White		
*C14*	F	75	White		
*C15*	F	83	White		
*C16*	F	71	White		
*C17*	F	78	White		
*C18*	M	58	White		
*C19*	M	60	Black		
*C20*	F	78	White		

* P identifies participant with dementia, C identifies carer.

**Table 2 ijerph-17-01544-t002:** Operational definitions of the constructs and examples of how they mediate behaviour change.

Construct	Operational Definition	Example of How the Construct Might Mediate Behaviour Change
*Characteristics of the person with dementia*	Characteristics of the person affecting behaviour change, which include personality, temperament and identity	Risk-takers might be more willing to challenge themselves in a physical activity programme than overly cautious subjects, thus potentially obtaining more positive outcomes
*Support*	Practical and emotional support from others (e.g., carer, therapist, society) which affects behaviour change	People might need an initial external push to initiate behaviour change, which may be provided by family members
*Expectations/goals*	Expectationsgoals around the behaviour, including benefits, barriers and facilitators	A person with dementia will sign up to an intervention delivering home-based physical exercise, if they believe that it will improve their health
*Carer(s)*	Any aspect, behaviour and attitude of the carer, which mediates behaviour change and maintenance	A carer might have risk-averse attitudes toward physical exercise and developing gate-keeping behaviour toward the person with dementia
*Progress*	Perceived or actual improvement in the person’s physical or mental health, following the behaviour	A person will find motivation to initiate/maintain behaviour change if they see progress/improvements
*Social opportunity*	Social contacts and networking opportunities (or lack thereof) granted through engaging in the behaviour	The opportunity for socialisation presented by a physical activity group in the community might encourage a person with dementia to sign up
*Self-efficacy*	Confidence in one’s ability to execute a given behaviour, including (perceived) physical, cognitive ability and competence	People with dementia might be reluctant to sign up for a walking group in the community, as they fear they might fall
*Capability*	One’s actual ability to perform a behaviour through essential skills, including (actual) physical, chronic conditions, cognitive ability, competence psychological/personal and social capability	People who have extensive memory impairment might struggle to remember the potential benefits that behaviour change might generate
*Activity/intervention characteristics*	Characteristics of the activity or intervention which influence participants’ engagement in it. They include how much the participant felt they are tailored to their needs, goal, preferences and aspirations, how helpful, enjoyable and challenging they are and how they fit into their routine	If a person finds an activity enjoyable, they will be more willing to engage
*Autonomy/control*	Being causal agents of one’s behaviour	A person will engage in behaviour change more easily when they have made an autonomous decision
*Physical infrastructure*	Environment and its characteristics, where the behaviour change occurs	A person with dementia who gives up their driving license might struggle to travel to an activity group organised in the community
*Personal history*	Personal history of a person, which affects present behaviour change	People who have been always very physically active are more motivated to engage in physical rehabilitation after hospitalisation
*Information/knowledge*	Information and knowledge that the person needs to change their behaviour	People who are informed about the benefits of behaviour change are more willing to initiate it
*Professional*	Any aspect, behaviour and attitude of the professional, which mediates behaviour change and maintenance	A person might be encouraged to exercise, if information on the benefits of exercising comes from a professional who is held in high esteem
*Personal beliefs*	The self-regulated mechanisms that the person uses in relation to initiation, adherence and withdrawal from behaviour change	A person might think that going to the gym could expose them to a higher risk of injury than spending more time at home

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
