# Peer review of "External Validation of the ‘PHYT in Dementia’, a Theoretical Model Promoting Physical Activity in People with Dementia"

_ijerph, 2020, doi:10.3390/ijerph17051544_

Round 1

Reviewer 1 Report

This paper examines support for the “PHYT in dementia” model describing the promotion of physical activity in people with dementia through the use of content analysis from patient and carer interviews. Overall, the paper was well written and clear. I provide just a few comments/suggestions below.  

Introduction

While I appreciate a streamlined introduction, this introduction may be too short. For example, it would be helpful to describe research that has shown that some of the constructs described in the PHYT-in-dementia model has empirical support. 

Methods

Additional information about the PrAISED study would be helpful – such as the N, and how the participants are recruited for the study.

Results

Pg. 4: How was the N = 40 decided upon?

Pg 5: The authors state “We found substantial inter-rater reliability (k = .65) between CDL and AB.”  Was this for the two interviews that AB reviewed? How were the two interviews selected (randomly?), and why only 2? Reliability is such an essential component in being able to make conclusions that the assessment of reliability may warrant the recoding of more than two interviews. Also, if the two coders did not agree on a designation, how was that disagreement resolved? 

Since the interviews lasted, on average, 48 minutes, how was the content divided up into segments to be categorized into one of the constructs?  More information would be helpful in terms of the coding process.

Table 2: It would be helpful to make a note either with a column, or a superscript, of which constructs are emerging and which are solidifying.

Author Response

Dear Reviewer, 

Thank you very much for your comments, which we feel have contributed to boost the quality of our manuscript.

We have carefully addressed each of your comments. For easy read, we have created a table, listing your comments and how we addressed them (see attachment below). We have also highlighted in yellow the relevant changes in the manuscript.

Thank you for your helpful comments.

The authors 

Reviewer 2 Report

General comments

The manuscript presents an initial validation of a theoretical model to promote physical activity in people with dementia. This is an important piece of work that will add value to future research in this area of developing potentially beneficial physical activity interventions for people with dementia. The use of Patient and Public Involvement in the research team is to be commended. Overall this is an important piece of work that was well-written and will help to better understand behaviour change in people with dementia.

Specific comments

Methods

Additional details are needed about the nature of the ethical approval for PrAISED 2 and whether the study in the manuscript was specifically mentioned.

Discussion

A stronger rationale for the changes in the model would be useful. For instance, why was the model changed with carers and health professionals grouped together? Why was the intervention and its characteristics removed from the model?

Author Response

Dear Reviewer

Thank you very much for your comments, which we are sure will boost the quality of our work. 

We have carefully addressed each of your comments and attached a table, with responses to how we addressed each of them. We have also highlighted in yellow the changes made in the text.

Thank you again for your helpful comments.

The authors 
